# Pharmacological Potential of Small Molecules for Treating Corneal Neovascularization

**DOI:** 10.3390/molecules25153468

**Published:** 2020-07-30

**Authors:** Zachary Barry, Bomina Park, Timothy W. Corson

**Affiliations:** 1Eugene and Marilyn Glick Eye Institute, Department of Ophthalmology, Indiana University School of Medicine, Indianapolis, IN 46202, USA; zbarry@iu.edu (Z.B.); parkbomi@iu.edu (B.P.); 2Department of Pharmacology and Toxicology, Indiana University School of Medicine, Indianapolis, IN 46202, USA

**Keywords:** corneal neovascularization, angiogenesis, inflammation, natural products, small molecules, natural molecules, drug discovery

## Abstract

Under healthy conditions, the cornea is an avascular structure which allows for transparency and optimal visual acuity. Its avascular nature is maintained by a balance of proangiogenic and antiangiogenic factors. An imbalance of these factors can result in abnormal blood vessel proliferation into the cornea. This corneal neovascularization (CoNV) can stem from a variety of insults including hypoxia and ocular surface inflammation caused by trauma, infection, chemical burns, and immunological diseases. CoNV threatens corneal transparency, resulting in permanent vision loss. Mainstay treatments of CoNV have partial efficacy and associated side effects, revealing the need for novel treatments. Numerous natural products and synthetic small molecules have shown potential in preclinical studies in vivo as antiangiogenic therapies for CoNV. Such small molecules include synthetic inhibitors of the vascular endothelial growth factor (VEGF) receptor and other tyrosine kinases, plus repurposed antimicrobials, as well as natural source-derived flavonoid and non-flavonoid phytochemicals, immunosuppressants, vitamins, and histone deacetylase inhibitors. They induce antiangiogenic and anti-inflammatory effects through inhibition of VEGF, NF-κB, and other growth factor receptor pathways. Here, we review the potential of small molecules, both synthetics and natural products, targeting these and other molecular mechanisms, as antiangiogenic agents in the treatment of CoNV.

## 1. Introduction

As the main refractive surface of the anterior aspect of the eye, the cornea plays a key role in optimal visual acuity (Figure 1). Corneal transparency is fundamental to its optical function and is possible due to its avascular structure. Under healthy conditions, the avascular nature of the cornea is maintained by a balance of proangiogenic and antiangiogenic factors. The cornea releases proangiogenic factors such as vascular endothelial growth factor (VEGF), platelet-derived growth factor (PDGF), basic fibroblast growth factor (b-FGF), and interleukins (ILs) that are sequestered and/or counterbalanced by local antiangiogenic factors that include angiostatin, pigment epithelium-derived factor (PEDF), and soluble vascular endothelial growth factor receptor 1 (sVEGFR-1) that maintain the corneal angiogenic privilege [1]. An imbalance of these factors allows for the abnormal proliferation of preexisting blood vessels (hemangiogenesis) and lymph vessels (lymphangiogenesis) into the corneal stroma, a process referred to as corneal neovascularization (CoNV).

CoNV arises due to a variety of insults including hypoxic injury and ocular surface inflammation due to trauma, infection, chemical burns, and immunological disease [2]. The increased vascular permeability of these new vessels leads to chronic corneal edema, lipid exudation, inflammation, and scar formation, thus compromising corneal transparency and potentially resulting in permanent vision loss [2]. The exact incidence and prevalence of CoNV are unknown, but CoNV is present in many cases of corneal disease, which is the 4^th^ leading cause of blindness globally after cataract, glaucoma and age-related macular degeneration according to the World Health Organization [3]. CoNV is also a common complication of corneal infections such as chlamydial infection, which is estimated to blind 4.9 million people due to scarring and vascularization [4]. 

Several medical and surgical options for treating CoNV exist. The mainstay treatment for CoNV is to suppress the inflammatory response with administration of topical steroids such as dexamethasone. Steroids suppress actively proliferating corneal vessels through their anti-inflammatory properties, which include inhibition of cell chemotaxis, proinflammatory cytokines, and prostaglandin synthesis [5]. However, steroids only provide incomplete suppression of CoNV [6] and are associated with major side effects such as corneal thinning, ocular hypertension, cataracts, and increased risk of infection [7]. Additional CoNV therapies consist of off-label use of anti-VEGF antibodies, such as bevacizumab, which has shown efficacy in treatment of other vascular ocular diseases such as age-related macular degeneration [8]. Bevacizumab has shown promising results in treating CoNV; however, partial efficacy, resistance, and side effects consisting of corneal thinning and reduced epithelial healing [9,10] have limited its use. Thus, there is need for safer and more effective therapies for CoNV. 

The search for novel drugs that block pathologic neovascularization has led to the study of numerous natural products and small-molecule inhibitors with varying biological mechanisms. Compared to large-molecule biologics, small molecules have the advantage of having various administration routes such as oral and topical (Figure 1). Additionally, they can potentially target multiple pathways and have favorable absorption, distribution, metabolism, excretion, and toxicity (ADMET) characteristics [11]. The efficacy of large-molecule biologics and other synthetic anti-VEGF therapies in the treatment of CoNV has been reviewed in the past [12]. Therefore, in this article, we focus on reviewing the potential of pure small molecules (excluding natural product extracts and other complex mixtures), both synthetic and natural products, targeting various molecular mechanisms, as antiangiogenic agents in the treatment of CoNV.

## 2. Models of CoNV

A majority of human CoNV cases are associated with ocular surface inflammation. Therefore, to study corneal hemangiogenesis, lymphangiogenesis, and tissue response to different therapies, multiple in vitro and in vivo models of CoNV have been developed. Here, we will introduce the preclinical cell-based assays and animal models of CoNV that will be referenced throughout the review. In vitro models of angiogenesis use cultured endothelial cells to test cell proliferation, migration, and tube formation in response to different compounds [13,14]. Human umbilical vein endothelial cells (HUVECs) are the most frequently used cell type for in vitro studies, although they are not a perfect surrogate of corneal endothelium. One model used to study angiogenesis in vivo is the corneal micropocket angiogenesis assay. This requires two adjacent micropocket incisions to be made in the mid cornea near the limbus for the implantation of a pellet of VEGF or b-FGF to stimulate angiogenesis while a pellet of the target antiangiogenic agent is inserted in the other micropocket [15]. This model is used to study the influence of specific molecules/proteins on angiogenesis [16] and is commonly used as a surrogate to study neovascularization in the context of other pathological processes such as cancer.

Chemical cauterization and suture placement models are the two most commonly used models for studying CoNV. Both of these model types have an inflammatory component, which mimics closely the complex nature of CoNV in human disease [16]. Chemical cauterization models induce CoNV by application of alkali (1N NaOH) or silver/potassium nitrate to the center of mouse, rat, or rabbit cornea for a short time followed by flushing with saline [17,18]. CoNV can be evaluated at 7–14 days after the procedure. In the suture-induced model, 7–0 silk or 10–0 nylon sutures are placed intrastromally in rabbit or rat/mouse cornea respectively. This results in CoNV response 7 days after surgery [16]. Finally, some studies use the corneal de-epithelialization model. In this model, scraping of the corneal epithelium from limbus to limbus is used to induce CoNV [19].

## 3. Synthetic Small Molecules

### 3.1. Tyrosine Kinase Inhibitors

Several proangiogenic factors such as VEGF, PDGF, and b-FGF mediate their angiogenic effect through interaction with a receptor tyrosine kinase (RTK). Synthetic small-molecule tyrosine kinase inhibitors (TKIs) interact with RTKs intracellularly and block downstream signaling pathways that stimulate angiogenesis [20]. Single-target TKIs and multitarget TKIs have shown promising antiangiogenic potential as both an anti-VEGF therapy and inhibition of multiple other angiogenic pathways (Table 1). 

Sunitinib is a multitarget TKI that potently inhibits VEGFRs, PDGFRs, c-KIT, and RET, and has been approved for the treatment of metastatic renal cell carcinoma (RCC) and gastrointestinal stromal tumors [21]. VEGF exerts its effect through interaction with vascular endothelial growth factor receptors (VEGFRs) that are RTKs. VEGFR1 and VEGFR2 are the major signal transducers of hemangiogenesis in CoNV, with VEGFR2 having the strongest hemangiogenic activity. VEGFR-3 on the other hand is the major contributor to lymphangiogenesis [22]. Oral administration of sunitinib to rats with thermal cauterization-induced CoNV showed significant reduction in lymphangiogenesis and hemangiogenesis. Immunostaining indicated decreased corneal F4/80+ cell infiltration and RT-PCR showed decreased *Vegfa* expression. This indicates that oral sunitinib was likely functioning through inhibition of VEGFR2 phosphorylation by macrophage secreted VEGF-A [23]. Other studies compared topical administration of sunitinib to topical bevacizumab. Both treatments were able to inhibit CoNV in a rabbit suture model. However, sunitinib was 3-fold more potent than bevacizumab, likely because of its inhibition of both the VEGF and PDGF pathways [24]. Further studies supported sunitinib’s efficacy over bevacizumab and indicated a greater inhibitory effect when administered topically rather than subconjunctivally [25]. While yellow deposits and iris staining were associated with topical administration and subconjunctival injections of sunitinib, no other toxicity or ocular side effects were observed in vivo.

AG 1296 is a single-target TKI that is selective for PDGF receptors (PDGFR). The dimeric ligand PDGF-BB interacts with the RTK, PDGFR-β resulting in downstream stimulation of VEGF and recruitment/proliferation of pericytes that contribute to vessel maturation [26]. Intraperitoneal injections of AG 1296 via an osmotic pump resulted in loss of pericytes and decreased vascularization by 21% in a murine de-epithelialization model. These changes were correlated with decreased expression of mRNAs for VEGF, PDGF, and angiopoietin 1/2. Similar changes were seen with the phosphatidylinositol 3-kinase (PI3K) inhibitors, wortmannin and LY294002, indicating that PI3K signaling is key to the downstream signaling of PDGF [27]. 

Vatalanib succinate (PTK787) is a potent oral multitarget TKI that is selective for all VEGFRs. ZK261991 is an oral VEGF TKI with selectivity for VEGFR2 and -3. Oral administration of vatalanib and ZK261991 resulted in significant reduction in lymphangiogenesis and hemangiogenesis and ZK261991 inhibited macrophage recruitment in a suture-induced CoNV model [28]. Therefore, similar to sunitinib, their anti-lymph/hemangiogenic effect is related to a combination of reduction in macrophage recruitment, which are major sources of prolymph/proangiogenic factors, and their direct effect on vascular endothelial cells [28].

Sorafenib is an orally active multitarget TKI with activity against VEGFRs, PDGFRs, c-RAF, FLT3, and c-KIT [29], that has been approved for treatment of hepatocellular carcinoma and advanced RCC [30]. Oral administration of sorafenib significantly reduced CoNV in a rat silver-nitrate model in a dose-dependent manner. RT-PCR and immunoblot showed reduced expression of corneal *Vegfr2* mRNA and phosphorylated ERK respectively in sorafenib treated rats compared to the control group. Therefore, sorafenib’s antiangiogenic effect is likely related to inhibition of VEGFR2 and ERK phosphorylation [31]. 

Semaxanib is a potent and selective TKI for VEGFR2 [32]. Intraperitoneal delivery of semaxanib significantly decreased new vessel formation in a murine silver-nitrate CoNV model [33]. A high occurrence of thromboembolic events has halted clinical development of semaxanib [34]; however, an earlier study showed intraperitoneal semaxanib to significantly reduce choroidal neovascularization as well [35], indicating that it may be beneficial in treating intraocular angiogenic diseases. Rivoceranib is another selective and potent VEGFR2 TKI that interferes with downstream angiogenic pathways. Topical application of rivoceranib in a murine alkali burn model demonstrated significant reduction in CNV area and reduction in lymph/hemangiogenesis that was equivalent to topical bevacizumab application [36].

Regorafenib is a multitarget TKI, inhibiting VEGFR-1, -2 and -3, PDGFR-β and FGFR, that has been approved for treatment of metastatic colorectal cancer [37]. Topical administration of regorafenib in a rat alkali burn CoNV model demonstrated decreased corneal VEGF expression and percentage of CoNV area that was comparable to topical dexamethasone 0.1% and bevacizumab [38]. 

Lapatinib is a multitarget TKI selective for human epidermal growth factor receptor 2 (HER2) and epidermal growth factor receptor (EGFR) used for treatment of HER2-positive breast cancer [39,40]. Oral administration of lapatinib reduced corneal epithelial and stromal VEGF expression, which correlated with decreased CoNV in a rat silver-nitrate CoNV model. Lapatinib was more effective at preventing CoNV than the large monoclonal antibody against HER2, trastuzumab [41].

Axitinib is a small multitarget TKI highly selective for VEGFRs and possibly PDGFRs. It is currently approved for treatment of RCC that has previously failed 1 year of systemic therapy [42]. Topical application of axitinib showed a dose-dependent inhibition of CoNV area and corneal stroma vascularization in a rabbit suture-induced CoNV model. Sunitinib tested using the same methodology showed similar reduction in CoNV with no significant difference in level of CoNV compared to axitinib [43].

Dovitinib is another multitarget TKI that inhibits VEGFR-1, -2, -3, and -4, FGFR-1, and -3, and PDGFR [44]. Studies of dovitinib’s antitumor effect have reported effective antiangiogenic properties. However, compared to topically administered bevacizumab, topically administered dovitinib was less effective in reducing CoNV in a rat silver-nitrate model [45]. However, dovitinib was administered at only one concentration (5 mg/mL), which may account for these findings. 

Many of the TKIs were studied as monotherapies against CoNV; however, the TKIs dovitinib, lapatinib, and sunitinib have also been tested in combination with other drugs. In an effort to broaden the multimechanistic approach to treating CoNV, sunitinib was combined with the tetracycline doxycycline and the natural product polyphenol hesperetin. When applied topically to a rat silver-nitrate model, sunitinib-hesperetin treatment was more effective at reducing CoNV than sunitinib alone and the sunitinib-doxycycline combination [46]. The combined anti-fibrotic effect of the hesperetin and antiangiogenic effect of sunitinib are believed to be the contributing factors to a greater therapeutic effect than monotherapy. However, not all TKI combination therapies showed greater effects than monotherapies. For example, the combination of dovitinib and bevacizumab was no more effective at reducing CoNV than bevacizumab monotherapy [45]. Additionally, lapatinib in combination with trastuzumab showed equivocal results to lapatinib monotherapy [41]. 

### 3.2. Repurposed Antimicrobials

The search for novel therapies has led to commercially available agents, such as antimicrobials, being assessed for their antiangiogenic properties (Table 2). Repurposing drugs has attracted much attention because it saves significant resources and time during drug development. Moreover, finding new indications for drugs for which safety has been already evaluated can bring great therapeutic benefits [47]. Tetracyclines are a group of small-molecule antibiotics that have been used safely in humans for years. Besides their antimicrobial activity, they prevent neovascularization by potently inhibiting collagenase and matrix metalloproteinase (MMP)-induced degradation of the extracellular matrix [48]. Oral administration of the semisynthetic tetracycline, doxycycline, significantly reduced CoNV compared to untreated controls in a murine alkali burn model [49]. In the same study, dexamethasone treated groups showed greater reduction in CoNV, but were associated with corneal ulceration unlike the doxycycline treated groups [49]. A similar study showed that topically administered 2% doxycycline reduced CoNV in a murine silver-nitrate model, and in combination with the steroid triamcinolone acetonide, it had a synergistic inhibitory effect [50]. The mechanism by which doxycycline inhibits CoNV is only partially due to MMP inhibition: MMP inhibitors, 1,10-phenanthroline and batimastat, had ~45% the inhibitory effect on VEGF-stimulated CoNV compared to doxycycline. In vitro studies showed that treatment with doxycycline significantly reduced PI3K activity and phosphorylation of Akt, indicating involvement of the PI3K/Akt-eNOS pathway in doxycycline-mediated inhibition of HUVEC proliferation. Together, the in vivo and in vitro studies support that the mechanism of doxycycline-mediated inhibition of angiogenesis occurs through a combination of MMP inhibition and the MMP-independent PI3K/Akt-eNOS pathway [51].

Minocycline is another semisynthetic tetracycline found to have antiangiogenic properties distinct from its bacteriostatic mechanism. Minocycline inhibits MMPs by upregulating tissue inhibitors of metalloproteinases-1 (TIMP-1), endogenous MMP inhibitors, similar to doxycycline but also downregulates the ERK1/2 and Akt pathways after VEGF stimulation [52]. When administered by intraperitoneal injection in a murine alkali burn CoNV model, minocycline significantly reduced CoNV, promoted corneal epithelial healing, and reduced corneal polymorphonuclear leukocytes (PMNs). Additionally, levels of corneal VEGF, VEGFR1, VEGFR2, b-FGF, IL-1β, IL-6, MMP-2, -9, and -13 were significantly reduced compared to control [53]. Therefore, minocycline’s effectiveness in treating CoNV is related to downregulation of angiogenic factors, inflammatory cytokines, and MMPs. 

Tigecycline is a newer and broader-spectrum tetracycline derived from minocycline [54]. CoNV was significantly reduced following both topical administration and subconjunctival injection of tigecycline in a rat silver-nitrate model, with subconjunctival injection being the most effective administration route [55]. It is likely that tigecycline acts by a similar mechanism to doxycycline and minocycline. However, its antiangiogenic mechanism and target have not yet been confirmed. 

Another antimicrobial agent to show potent antiangiogenic activity is the antifungal **itraconazole**. It prevents angiogenesis through inhibition of cholesterol biosynthesis which is needed for endothelial cell proliferation and capillary formation [56]. Topical, subconjunctival, and intraperitoneal administration of itraconazole were studied in a rat silver-nitrate model. Biomicroscopic examination after treatment showed significant inhibition of CoNV in all treatment groups with topical and subconjunctival administration being the most effective [57]. Such antiangiogenic efficacy of itraconazole indicates targeting endothelial cell metabolism such as cholesterol synthesis as a potential therapeutic strategy to treat CoNV.

The antimalarial, dihydroartemisinin, is a semisynthetic derivative of artemisinin, which is isolated from the Chinese herb, *Artemisia annua* [58]. In addition to its antimalarial properties, in vitro and in vivo studies have shown that dihydroartemisinin can inhibit angiogenesis. Dihydroartemisinin treatment induced apoptosis and reduced expression of VEGF in HUVECs [59]. Topical administration of dihydroartemisinin to rats with suture-induced CoNV significantly decreased CoNV area and corneal VEGF, VEGFR2 phosphorylation, ERK1/2 and p38 expression [60]. This suggests that the ERK1/2 and p38 pathways are partially involved in dihydroartemisinin’s antiangiogenic mechanism. 

### 3.3. Other Synthetics

The pathogenesis of CoNV is a complex process involving multiple mechanisms and angiogenic growth factors. There are various synthetic small molecules other than TKIs and repurposed antiangiogenic antimicrobials that have undergone preclinical testing for CoNV treatment. These synthetic molecules induce their antiangiogenic properties through inhibition of integrins, ROS, and inflammation (Table 3). 

Endothelial cell activation, survival, migration, and adhesion to extracellular matrix are essential steps in the process of angiogenesis. Many of those steps are regulated by transmembrane glycoproteins called integrins. The integrin α_5_β_1_ was found to impact blood vessel formation and maturation [61]. Inhibition of α_5_β_1_ reduced pathological neovascularization in tumor animal models. Osmotic pump intraperitoneal delivery of the anti-integrin α_5_β_1_ small molecule, JSM5562, significantly regressed development of CoNV in a murine alkali burn model [62]. This effect was likely a result of JSM5562 impairing endothelial cell migration, adhesion, and tube formation. However, further studies are needed to assess the exact mechanism that the integrin α_5_β_1_ plays in the inflammatory context.

Similarly to JSM5562, the imidazole-based alkaloid derivative LCB54–0009 also inhibits endothelial cell capillary-like tube formation. LCB54–0009 exhibits antiangiogenic and antioxidant activity associated with hypoxia-inducible factors (HIFs) in preclinical in vitro and in vivo studies. Under low oxygen levels HIFs induce the transcription of proangiogenic and inflammatory molecules [63]. Subconjunctival LCB54–0009 in the murine suture model showed reduction in CoNV and inflammation. LCB54–0009 treatment of hypoxic and VEGF-stimulated HUVECs showed significant inhibition of tube formation and levels of HIF-1α, angiopoietin-2, and phosphorylated VEGFR2 [64]. Therefore, the likely mechanism of LCB54–0009′s antiangiogenic effect is through the inhibition of ROS-mediated signaling cascades, resulting in the inhibition of the HIF-1α, NF-κB, and VEGF/VEGFR2 signaling pathways [64]. 

*N*-acetyl-l-cysteine (NAC) is another synthetic small molecule that was found to inhibit CoNV through its antioxidant properties. ROS activate the transcription factor NF-κB, which induces the expression of inflammatory cytokines such as VEGF, monocyte chemoattractant protein (MCP)-1, IL-1β, and TNF-α [65]. NAC downregulates the NF-κB pathway. To investigate the role of ROS in an angiogenic response, intraperitoneal NAC was administered in a murine alkali burn model. Pretreatment starting 3 days before corneal injury showed significant reduction in CoNV and reduced expression of VEGF, MCP-1, and NF-κB phosphorylation [66]. While pretreatment with antioxidants is not clinically feasible, it does indicate that antioxidants such as NAC may protect the cornea from pathological angiogenesis through its ROS-blocking effects. 

IMD0354 is a synthetic small molecule that also interferes with NF-κB signaling. The activation of the NF-κB pathway is regulated by the IKK complex of two kinases, IKK1 and IKK2. IMD0354 is a non-ATP binding, competitive, selective IKK2 inhibitor that has shown inhibitory effects on VEGF expression in murine diabetic retinopathy models [67]. In rat suture-induced CoNV, systemic IMD0354 significantly suppressed CoNV, decreased inflammatory cell infiltration, expression of inflammatory chemokines (CCL2, CXCL5, Cxcr2), and attenuated expression of angiogenic factors (VEGF) and inflammatory mediators (TNF-α) [68]. In vitro studies support the antiangiogenic potential of IMD0354: it inhibited HUVEC migration and tube formation and downregulated VEGF-A expression [68]. The results of this testing underscore the importance of NF-κB signaling in the mechanism of CoNV. 

Due to the association of inflammation and CoNV, the role of the inflammatory mediator substance P has also been tested in CoNV murine models. Substance P mainly exerts its proinflammatory effect through the interaction with neurokinin 1 receptor (NK1R) [69], and substance P levels were found to be increased in murine corneas following alkali burn injury [70]. Therefore, the NK1R antagonist, lanepitant, was applied topically and subconjunctivally in murine alkali burn and suture-induced CoNV models. Significant reduction in CoNV, corneal substance P expression, and leukocyte infiltration were seen in the alkali burn model with subconjunctival and topical administration and only with subconjunctival injection in the suture-induced model [71]. Limited drug penetration with topical application in the suture model may have accounted for these disparate results. Overall, the antiangiogenic effects of lanepitant suggest that inhibition of NK1R may offer therapeutic effects in the treatment of inflammation-induced CoNV. 

Inflammation in CoNV is also related to the overexpression of proinflammatory chemokines that are mediators of angiogenesis. Chemokines bind to specific G-protein coupled receptors to induce cell migration and activation. Multiple chemokine receptors, such as CCR3 and CXCR-4, have been shown to be upregulated in ocular neovascularization models [72,73]. The CCR3 antagonist **SB-328437** applied topically to the corneas of alkali burned mice successfully suppressed CoNV and reduced intracorneal levels of mRNAs for MCP-1 and -3, which promote expression of VEGF and chemotaxis of macrophages [74]. This suggests that CCR3 signaling may be involved in the development of CoNV by its interaction with infiltrating inflammatory cells rather than directly with vascular endothelial cells. However, further studies are needed to determine the exact mechanism of CCR3 signaling on CoNV inhibition. 

The chemokine receptor CXCR-4 has also been shown to be involved in CoNV. CXCR-4 is involved in the recruitment of adult endothelial progenitor cells (EPC) which are of bone marrow origin and contribute to physiologic and pathologic neovascularization [75]. The exact mechanism of CXCR-4 in regulating angiogenesis is unclear but chemokine stromal cell-derived factor-1 (SDF-1) engages with CXCR-4 expressed by vascular cells and promotes mobilization of proangiogenic hematopoietic cells that express CXCR-4 and VEGFRs, thereby inducing revascularization of ischemic tissues [76]. Interestingly, previous studies indicated that a specific antagonist of CXCR-4, AMD3100, has dual functions, improving blood circulation in ischemic tissue by promoting mobilization of EPCs from bone marrow into peripheral blood in myocardial infarction patients, while also inhibiting angiogenesis by targeting CXCR-4. To investigate whether EPCs are involved in CoNV and whether blockade of the SDF-1/CXCR-4 axis affects CoNV formation, AMD3100 was injected subconjunctivally in a murine alkali burn model and was shown to significantly inhibit CoNV, corneal inflammation and number of inflammatory cells, and downregulate VEGFR2 expression. However, systemic delivery of AMD3100 by intraperitoneal injection did not have a significant effect on either corneal inflammation or CoNV. Although it needs to be further studied, it is speculated that AMD3100 in the systemic blood supply may accelerate mobilization of bone marrow EPCs into ocular circulation, promoting EPC localizing in the lesions caused by corneal injury. This demonstrates how different routes of administration can result in conflicting therapeutic effects and highlights the importance of local delivery of ocular therapeutics [77]. 

Corneal fibrosis is commonly associated with the development of CoNV due to the main etiologies involving inflammation. A family of animal lectins that bind β-galactosides named galectins are involved in CoNV as well as fibrosis. Galectin-3 is an important modulator of the VEGF/VEGFR2 signaling pathway and has increased corneal expression in states of inflammation [78]. The galectin-3 inhibitor, 33-DFTG (TD139) showed antiangiogenic effects by attenuating VEGF-induced HUVEC migration and sprouting [79]. Due to galectin-3′s proangiogenic function, subconjunctival injections of 33-DFTG were also administered to murine silver-nitrate and alkali burn models, which significantly suppressed CoNV, corneal fibrosis, and expression of corneal α-smooth muscle actin (SMA) [79]. An eye drop formulation of 33-DFTG was also effective at reducing CoNV. This suggests that inhibition of galectin-3 could have a beneficial role in the treatment of CoNV and corneal fibrosis. 

Finally, TNP-470 is a synthetic analogue of fumagillin, an antibiotic compound secreted by *Aspergillus fumigatus*. TNP-470 has potent antiangiogenic properties through the inhibition of type 2 methionine aminopeptidase [80,81,82]. This activity was supported by in vitro testing that revealed that TNP-470 significantly inhibited proliferation of b-FGF and VEGF induced bovine capillary endothelial cells. Additionally, both systemic and topical administration to a murine alkali burn model significantly reduced CoNV and VEGF expression [17]. TNP-470 has therapeutic potential as either a monotherapy or in combination with anti-inflammatory drugs. However, more studies are required to find an optimal topical dose and to assess for longer-term ocular toxicity.

## 4. Natural Products

### 4.1. Polyphenols: Flavonoids

Natural products derived from plants, animals, and microorganisms have played an essential role in the history of medicine and drug discovery. Natural products have immense structural and chemical diversity that has allowed for the development of a broad spectrum of pharmaceuticals [83]. Polyphenols make up the largest group of bioactive molecules that are commonly found in plant-based foods. The polyphenols are classified into two main groups: flavonoids and non-flavonoids. More than 4000 flavonoids have been identified and can be further subdivided into various structural subclasses such as flavanols, flavones, isoflavones, flavonols, and chalcones that will be of interest in this review [84]. Due to their polypharmacology, natural products have been recognized for their potential multitarget therapeutic effects and have been studied in a variety of disease states such as CoNV (Table 4). 

The flavanol, epigallocatechin gallate (EGCG), is the major secondary metabolite present at high levels in green tea. EGCG has been shown to inhibit multiple downstream signaling pathways and therefore has been studied for its chemopreventative, antioxidant, anti-inflammatory, and antiangiogenic properties [85,86,87,88]. Topical application of EGCG to suture-induced CoNV in rabbits showed significant reduction in CoNV surface area and expression of *Vegfa* mRNA and cyclooxygenase-2 (COX-2) protein [89]. COX-2 is a proangiogenic protein through modulation of VEGF ligand and receptors [90]. Therefore, EGCG’s inhibitory effect in CoNV is likely related to suppression of VEGF and COX-2 mediated angiogenesis. In an attempt to maximize the bioavailability of EGCG eye drops, arginine-glycine-aspartic acid peptide-hyaluronic acid-conjugate complex-coated, gelatin/EGCG self-assembling nanoparticles (GEH-RGD NPs) were synthesized. In a murine alkali burn model, fewer and thinner corneal vessels were observed after treatment with GEH-RGD NP eye drops compared to mice treated with free EGCG solution eye drops [91]. Topical application of EGCG has been shown to be effective for the treatment of CoNV, however EGCG’s underlying mechanism of CoNV inhibition has yet to be identified. 

Kaempferol belongs to the flavonol subclass and is one of the most common dietary flavonoids. In vitro studies of kaempferol have demonstrated its ability to inhibit angiogenesis, VEGF expression, inflammation, and vascular cell migration [92,93]. To test its ability to inhibit CoNV, gelatin nanoparticles with kaempferol encapsulation (GNP-KA) were formulated to increase bioavailability and were administered to HUVECs and topically to a murine silver/potassium nitrate model. GNP-KA significantly suppressed cell migration of HUVECs and had less vessel growth into the cornea compared to kaempferol solution by reducing corneal MMP and VEGF [94]. These results indicate that topical application of kaempferol in nanoparticle formulation could be a viable candidate for treatment of CoNV. 

Isoliquiritigenin is a chalcone flavonoid isolated from the root of licorice (*Glycyrrhiza uralensis* and *G. glabra*). Isoliquiritigenin has diverse pharmacological properties: it is anti-inflammatory, anti-viral, anti-microbial, antioxidant, anticancer, immunomodulatory, hepatoprotective, and cardioprotective [95]. Topical administration of isoliquiritigenin in a murine silver-nitrate CoNV model suppressed NV in a dose-dependent manner. This study also showed isoliquiritigenin reduced VEGF and increased antiangiogenic factor PEDF in VEGF stimulated HUVECs [96]. This suggests that the potential role of isoliquiritigenin in suppressing angiogenesis is through restoring the balance of pro and antiangiogenic factors, however, this still needs to be confirmed in corneal tissue. 

Diets containing flavonoids and isoflavonoids have been of interest for novel therapies due to their potential of regulating angiogenesis. The flavones fisetin and luteolin and isoflavone genistein are commonly found in many fruits and vegetables. To evaluate their effect on CoNV, fisetin, luteolin, and genistein were applied topically to rabbits with CoNV induced by b-FGF pellets implanted intrastromally. All three substances significantly reduced CoNV, with fisetin having the strongest effect followed by genistein and luteolin [97]. In murine xenografts of breast carcinoma cells, genistein downregulated VEGF, TGF-α, MMP-9, and upregulated TIMP-1 [98], and therefore, CoNV inhibition by genistein, fisetin, and luteolin in this study may involve similar mechanisms. However, the exact mechanism by which these flavonoids and isoflavonoids enact their antiangiogenic effect in CoNV has yet to be discovered. 

Naringenin is a flavanone that is abundant in vegetables and citrus fruits. Previous studies have demonstrated that naringenin has anti-inflammatory properties related to the downregulation of NF-κB activity and reduced production of inflammatory cytokines IL-1β and IL-6 [99,100]. Additionally, a rat choroidal neovascularization model showed that naringenin’s antiangiogenic potential is linked to its anti-inflammatory properties and downregulation of VEGF and COX-2 [101]. When administered topically in a murine alkali burn CoNV model, naringenin reduced CoNV, inhibited corneal recruitment of neutrophils and macrophages, and decreased production of cytokines (IL-1β and IL-6) and *Vegfa*, *Pdgf*, and *Mmp14* mRNA expression [102]. These results suggest that in addition to reducing proangiogenic factors, the anti-inflammatory properties of naringenin play a similar role in inhibiting corneal angiogenesis as seen in earlier studies on choroidal neovascularization [101]. 

While several flavonoids have shown favorable results for treating CoNV, this is not the case for every study. The flavonol quercetin and the coumarin esculetin have inhibitory activities against lipoxygenase (LOX) which is one of the key enzymes of the arachidonic acid (AA) metabolic pathway. The AA metabolic pathways can give rise to proinflammatory molecules [103] and thus they are well-known therapeutic targets of anti-inflammatory agents such as steroids and COX inhibitors that block the COX and LOX pathways at the first step of the AA pathway. Steroids and COX inhibitors have shown the ability to inhibit CoNV in inflammatory CoNV models [18,104]. To further assess the involvement of these pathways in CoNV, quercetin and esculetin were studied due to their ability to inhibit LOX [105,106]. However, topical application of 1% esculetin and quercetin in a rat silver-nitrate CoNV model was not able to significantly reduce CoNV compared to control [104].

### 4.2. Non-Flavonoid Phytochemicals

Plants produce a vast group of organic compounds other than flavonoids referred to as phytochemicals. Some of these natural products have been of special interest in the search for alternative therapies for treating CoNV due to their diverse and complex chemical properties. Therefore, plant phytochemicals such as non-flavonoid polyphenols, a steroidal lactone, and a sesquiterpene lactone have been studied in preclinical CoNV studies (Table 5). 

Curcumin, the major curcuminoid polyphenol, is isolated from the turmeric plant. Earlier studies found curcumin to possess antitumor effects that are mediated by antiangiogenic, as well as potent antioxidant, anti-inflammatory properties [107,108,109]. In vivo and in vitro studies showed that curcumin also has the ability to inhibit CoNV. Topical administration of curcumin significantly reduced CoNV and suppressed corneal VEGF expression in a rabbit suture model [110]. In vitro, curcumin successfully suppressed VEGF-stimulated HUVEC migration while inhibiting NF-κB activation [111]. Therefore, a possible mechanism of curcumin mediated CoNV inhibition is through downregulation of VEGF by inhibition of NF-κB activity. These findings were supported by curcumin nanoparticle eye drops significantly inhibiting corneal NF-κB, CoNV, and expression of inflammatory cytokines plus VEGF in a rat silver-nitrate model [112]. While the inhibition of several angiogenic pathways supports a multitarget approach of curcumin-induced CoNV inhibition, the exact antiangiogenic mechanism and target are currently unknown. These studies indicate curcumin has potential as a therapy for CoNV, but it is worth mentioning its drawbacks. The shortcomings associated with curcumin include poor pharmacokinetics/pharmacodynamic properties, unclear mechanism, low efficacy in several disease models, and toxic effects under certain testing conditions [113]. 

Resveratrol is another non-flavonoid polyphenol present in red wine and other grape products. Previous studies have shown resveratrol to have chemopreventative properties through inhibition of tumor initiation, promotion, and progression [114]. Oral resveratrol significantly suppressed murine FGF-2 and VEGF-micropocket induced CoNV, suggesting disruption of blood vessel growth as a part of resveratrol’s antitumor effect [115]. Resveratrol’s antiangiogenic effects sparked interest in its potential as a therapy for CoNV. However, in a rabbit inflammatory alkali burn CoNV model, subconjunctival injection of 1% resveratrol did not significantly reduce CoNV compared to control [116]. Despite the negative results of subconjunctival resveratrol at this dose, further studies are needed to assess whether an optimal dose and route of administration of resveratrol can inhibit CoNV.

Another phytochemical with diverse pharmacological properties is the steroidal lactone withaferin A, which is extracted from the root of the medicinal plant *Withania somnifera*. Withaferin A has anti-inflammatory, immunosuppressive, antitumor, and antiangiogenic activity [117]. Its antiangiogenic activity is due to targeting and downregulation of the intermediate filament protein vimentin. This is supported by intraperitoneal injection of withaferin A in a deepithelialization model of CoNV markedly suppressing CoNV in wild-type mice while only marginally attenuating CoNV in vimentin-null mice [118]. The use of vimentin-null mice in this study validates that withaferin A’s mechanism of CoNV inhibition is largely mediated through its interaction with vimentin, adding an extra level of rigor to this work. 

The sesquiterpene lactone xanthatin is the major bioactive compound isolated from the leaves of *Xanthium sibiricum*. Xanthatin has beneficial biological activities such as antitumor, antifungal, and antiplasmodial effects [119,120,121]. Xanthatin was also shown to have antiangiogenic properties through the inhibition of the VEGFR2 signaling pathways [122]. Xanthatin’s antiangiogenic effect and mechanism in CoNV were further evaluated with in vitro and in vivo studies. Xanthatin inhibited the migration and lumen-forming ability of VEGF-treated HUVECs in a concentration-dependent manner and significantly decreased expression of phosphorylated (p-)VEGFR2, p-STAT3, p-PI3K, and p-Akt [123]. Topical xanthatin eye drops were used for the treatment of CoNV in a rat alkali burn model, and suppressed CoNV area, lowered VEGF, and raised PEDF protein levels [123]. Together, these results indicate that xanthatin likely inhibits CoNV through regulating the VEGFR2-mediated STAT3/PI3K/Akt signaling pathways. 

Triptolide is a phytochemical extracted from *Tripterygium wilfordii* Hook F and is a key ingredient of Chinese herbal medicine. It has diverse pharmaceutical activities such as anti-inflammatory, antiproliferative, proapoptotic, and immunosuppressive properties [124,125]. To evaluate its effect on angiogenesis, it was tested in vitro and in vivo. Triptolide inhibited rat aortic endothelial cell migration and tube formation and significantly suppressed CoNV and VEGF expression in a murine alkali burn model [126]. While these results indicate that triptolide may have clinical indications for the treatment of CoNV, further studies are needed to assess its exact mechanism of action, optimal dose, and potential toxic side effects with long term use. 

Thymoquinone is another biologically active compound isolated from the volatile oil of black seed (*Nigella sativa*). It has anti-inflammatory properties through the inhibition of the cyclooxygenase and lipoxygenase pathways [127], as well as antioxidant and antineoplastic effects both in vitro and in vivo [128,129,130]. Therefore, it was studied for its inhibitory effect on CoNV. Topical application of thymoquinone reduced CoNV in a dose-dependent manner in a rat silver-nitrate model and was also found to be as effective as topical triamcinolone in reducing CoNV [131]. Further studies are needed to determine thymoquinone’s exact mechanism of inducing CoNV inhibition, but it is believed to be mediated by its antioxidant or anti-inflammatory properties. 

**Glycyrrhizin** is another major active constituent of licorice (*G. glabra*), and has shown anti-inflammatory and anticancer activity [132]. Topical glycyrrhizin was applied to rabbit corneas following alkali burn. Glycyrrhizin treatment resulted in considerable decrease in CoNV but was less effective than *G. glabra* extract, which contained several chemical constituents [133]. This suggests that licorice root extract has potential as a CoNV therapy, but further research is needed to identify the main components responsible for its antiangiogenic effect in CoNV.

### 4.3. Immunosuppressants

There are various causes of CoNV, with a majority of these etiologies associated with an inflammatory response. Due to the limited efficacy and side effect profile of the standard-of-care steroids, the search for novel therapies that target the inflammatory response continues. Several natural metabolites produced by fungi and bacteria are effective immunosuppressive agents that have shown promising therapeutic potential for CoNV (Table 6). 

Tetramethylpyrazine (TMP) is the major bioactive component of the traditional Chinese medicine, chuanxiong (*Ligusticum striatum*). In China, chuanxiong is prescribed for cancer, autoimmune, and inflammatory diseases [134,135]. Previous in vitro studies identified that TMP significantly downregulates CXCR4 expression in HUVECs [136]. Topical application of TMP significantly suppressed CoNV, macrophage aggregation, and corneal CXCR4 expression in a murine alkali burn model and was as effective as the immunosuppressant tacrolimus (see below) [137]. These results suggest that the mechanism of TMP-mediated inhibition of CoNV is likely through regulation of CXCR4 and TMP eye drops could be a potential agent for CoNV treatment.

Synthetic immunomodulators have also been studied as novel therapies for CoNV. Methotrexate is a synthetic antimetabolic substance that blocks DNA and RNA synthesis through antagonism of folic acid [138]. It is used in the treatment of autoimmune diseases, however, a study identified that systemic administration decreased cerebral blood flow [139]. Therefore, methotrexate was examined for its antiangiogenic potential in a rabbit suture-induced model of CoNV. Topical and subconjunctival injection of methotrexate significant decreased CoNV area and expression of VEGF and IL-6 in treated corneas [140]. A similar antiangiogenic effect was seen with topical methotrexate in a rabbit corneal pocket model [141]. These studies indicate methotrexate’s efficacy in treating CoNV, however, further studies are needed to determine safe and effective doses. 

Thalidomide contains a single stereogenic carbon and therefore it exists in (*R*)- and (*S*)-enantiomers. These enantiomers have different biological properties and only the (*S*)-enantiomer is teratogenic [142]. However, the drug was initially marketed as a racemate and its use in humans has been limited due to teratogenic effects [143,144]. Since then, thalidomide has gone through a significant transformation from a notorious sedative causing birth defects to a compound that is now again garnering a clinical interest due its pharmacological effects against numerous pathological processes [145]. Thalidomide is immunomodulatory and anti-inflammatory, and it potently inhibits angiogenesis caused by TNF-α and FGF-2 [143]. **CC-3052** is an analogue of thalidomide with the same antiangiogenic effects and greater immunomodulatory activity, but is also nontoxic, nonmutagenic, and nonteratogenic [146]. Work in a rabbit suture-induced CoNV model showed that topical and subconjunctival injections of CC-3052 were effective at decreasing CoNV area and expression of mRNAs for VEGF and TNF-α. Additionally, on histopathological analysis, both topical and subconjunctival administration reduced inflammation intensity, fibroblast activity, and neovascularization [147]. Similar to methotrexate, CC-3052′s anti-inflammatory effect reduced CoNV; however, optimal dose and administration route have yet to be determined. Another analogue of thalidomide, DAID, has antiproliferative and antimitotic activities in vitro [148]. DAID was applied topically to alkali burn mouse corneas and reduced CoNV and attenuated the overexpression of VEGF [149]. DAID has potential as a noninvasive therapy for CoNV; however, more studies are needed to determine how it suppresses VEGF. LASSBio-596 is another hybrid of a thalidomide derivative and an arylsulfonamide, without the teratogenic profile of thalidomide, that displays anti-inflammatory and immunomodulatory properties [150,151]. Topical application of LASSBio-596 to a rabbit alkali burn model significantly reduced CoNV [152]. LASSBio-596 was not as effective at reducing CoNV as dexamethasone, but due to its ability to suppress angiogenesis further studies may be beneficial to better understand its effect on the process of angiogenesis.

#### Macrolides

In addition to the immunosuppressants discussed above, the macrolide immunosuppressants have also been tested for their potential to treat CoNV. Cyclosporine A is a cyclic nonribosomal peptide secondary metabolite of the fungal genus *Tolypocladium* and is an immunosuppressive drug use to treat various autoimmune and inflammatory conditions through suppression of T-lymphocyte functions. In a rat silver-nitrate model, topical and subcutaneous injections of cyclosporine A reduced blood vessel formation [153]. A similar study assessed the inhibitory effects of topical cyclosporine A 0.05% on immune-mediated rabbit CoNV and compared its efficacy to topical dexamethasone and bevacizumab. Cyclosporine A significantly suppressed CoNV and was more effective than bevacizumab, but was not as effective as dexamethasone [154]. Long-term systemic administration of cyclosporine A often leads to toxic damage of various organs, and therefore, more focus has been placed on its topical use [155]. The unique clinical accessibility of the eye allows targeted drug delivery by routes such as topical eye drops, therefore minimizing unwanted systemic side effects. To better improve bioavailability of ocular cyclosporine A it was incorporated into nanofibers and applied to rabbit corneas following alkali burn. The corneas treated with cyclosporine A-loaded nanofibers showed strongly suppressed CoNV and restoration of corneal transparency, had significant reduction in CD3(+) cells and proinflammatory cytokines, and decreased expression of MMP-9, inducible nitric oxide synthase, and VEGF [156]. While the exact mechanism of cyclosporine A-induced inhibition of CoNV has yet to be identified, topical application shows promising therapeutic potential.

Rapamycin (sirolimus) is a macrolide product of *Streptomyces hygroscopicus* that inhibits mammalian target of rapamycin (mTOR) making it an effective immunosuppressant for treatment of allograft rejection and several cancers [157,158]. In vitro assessment of its angiogenic activity revealed that rapamycin strongly inhibited HUVEC proliferation and migration but did not cause cytotoxicity. An in vivo study with the murine alkali burn model showed reduced CoNV and reduced expression of proinflammatory factors (substance P, VEGF, TNF-α, TGF-β and IL-6) by topical and intraperitoneal injections of rapamycin [159,160]. Additionally, the rapamycin synthetic derivative, everolimus, has shown similar results on CoNV. Topically applied everolimus in a murine silver-nitrate model significantly reduced CoNV and levels of mRNAs for VEGFR2 and ERK 1/2, and appears to be more effective than sunitinib [161]. These results reveal the antiangiogenic potential of rapamycin and its derivatives, but future studies are needed to determine the safest and most effective route of administration.

Tacrolimus (FK506), a macrolide isolated from *Streptomyces tsukubaensis*, is an immunosuppressant 100-fold more effective than cyclosporine A that is commonly used to prevent human organ transplant rejection [162]. Topical and intraperitoneal tacrolimus were administered to rats with silver-nitrate induced CoNV, resulting in decreased CoNV area and decreased intensity of VEGF immunostaining compared to control groups [163]. Due to its effect on VEGF, subconjunctival and topical tacrolimus were compared to bevacizumab in a rabbit suture CoNV model, revealing that subconjunctival injections of tacrolimus were similar to bevacizumab in reducing CoNV area. Additionally, both groups showed reduced levels of VEGF, inflammatory cytokines (TNF-α, IL-1β, and MCP-1), and infiltration of F4/80+ inflammatory cells [164]. While the exact antiangiogenic mechanism of tacrolimus remains unknown, these results suggest that a combination of immunosuppression and inhibition of the VEGF pathways are involved in CoNV inhibition. 

### 4.4. Vitamins and Photoactivatable Compounds

Water- and fat-soluble vitamins are another source of dietary natural products that are essential for human growth and health. In vitro and in vivo studies also indicate that the vitamins ascorbic acid, riboflavin, and vitamin D_3_ possess antiangiogenic properties (Table 7). **Ascorbic acid** (vitamin C) has been shown to inhibit MMP-9 and VEGF expression in models of tumor growth [165]. When applied to a silver-nitrate CoNV model, topical ascorbic acid was found to reduce neovascularization in a dose-dependent manner [166]. These results were further supported by the suppression of CoNV and downregulation of VEGF and MMP-9 by topical ascorbic acid administration in a rabbit suture model [167]. 

Riboflavin (vitamin B_2_) is another water-soluble vitamin with antiangiogenic potential when combined with UV light. This process is referred to as corneal crosslinking, and it is used to treat other ocular diseases such as progressive keratoconus, corneal ulcers, pellucid marginal degeneration, corneal melting, and iatrogenic keratectasia after laser surgery [168,169]. Topical riboflavin is photoactivated by UVA light, resulting in the release of reactive oxygen radicals which cause apoptosis [170]. The corneas of mice with suture-induced CoNV received corneal crosslinking treatment with riboflavin and UVA rays which resulted in regression of preexisting blood and lymphatic vessels via induction of apoptosis of vascular endothelial cells. Additionally, there was a significant reduction in macrophages and CD45+ cells in inflamed corneas [171]. This is similar to the results seen in photodynamic therapy using the photosensitizer **verteporfin**. Verteporfin is used for treatment of angiogenic diseases such as certain cancers and subfoveal choroidal neovascularization [172,173]. Although it is not a vitamin, it is a synthetic small molecule that when injected intravenously and combined with photodynamic therapy significantly reduced preexisting blood and lymphatic vessels in the corneas of suture-induced mice [174]. The ability of these photoactivated therapies to reduce CoNV and inflammatory cells indicates the potential combination of light and riboflavin or verteporfin in treating preexisting CoNV and for decreasing graft rejection after corneal transplant in high risk eyes. 

The active form of vitamin D_3_, 1α,25-dihydroxyvitamin D_3_ (1α,25[OH]_2_D_3_), is a hormone involved in the regulation of calcium homeostasis. However, 1α,25[OH]_2_D_3_ also has antiangiogenic effects in a transgenic murine retinoblastoma model [175] and other angiogenesis models [176]. To test its effect on CoNV, it was applied topically to mouse corneas with suture-induced CoNV. 1α,25[OH]_2_D_3_ reduced CoNV and corneal inflammation by inhibiting intracorneal migration of antigen presenting cells involved in initiating the immune response called Langerhans cells [177]. These results suggest that topical vitamin D_3_ has potential as a treatment for CoNV, but further testing is needed to rule out toxicity of 1α,25[OH]_2_D_3_ on corneal epithelial cells. 

### 4.5. HDAC Inhibitors

Histone deacetylases (HDACs) are a family of enzymes that regulate gene transcription through the modification of histone and nonhistone proteins by acetylation. HDACs play a role in cell proliferation and survival and are essential for angiogenesis through the regulation of VEGF expression [178]. Largazole is a natural macrocyclic depsipeptide isolated from the marine cyanobacterium *Symploca* species that selectively inhibits class I HDACs [179]. Topical application of largazole to murine alkali burn-injured corneas attenuated CoNV by downregulating the proangiogenic factors VEGF, b-FGF, TGF-β1, and EGF and upregulating antiangiogenic factors thrombospondin-1 (Tsp-1), Tsp-2, and ADAMTS-1. Furthermore, largazole inhibited migration, proliferation, and tube formation of a human dermal microvascular endothelial cell line (HEMC-1) [180]. Additional studies on the potent synthetic HDAC inhibitor, **vorinostat** (suberoylanilide hydroxamic acid; SAHA), showed similar results to largazole in vivo and in vitro [181]. Topical application of vorinostat inhibited CoNV in a murine alkali burn model through the inhibition of hemangiogenesis, lymphangiogenesis, and inflammation [182]. The exact mechanism of CoNV inhibition by vorinostat remains unknown, however its overall effect on CoNV was comparable to current steroid therapies. Together, these studies indicate that both natural and synthetic HDAC inhibitors have potential as therapeutic drugs for CoNV associated with inflammation (Table 8). 

## 5. Discussion/Future Directions

CoNV is a common sequela of numerous corneal insults from hypoxia and inflammatory conditions. The mechanisms involved in the induction of CoNV are complex and are regulated by a multitude of pro and antiangiogenic factors in the cornea. A majority of CoNV cases involve inflammatory conditions and VEGF is arguably one of the main factors involved in the development of NV. However, despite the usefulness of corticosteroids and anti-VEGF agents in suppressing CoNV their partial efficacy and side effect profile reveal the need for novel therapies. This review highlighted a growing collection of synthetic and natural small-molecule inhibitors of various molecular mechanisms that showed promising antiangiogenic potential against CoNV in preclinical in vitro and in vivo studies. The next step for many of these compounds would be to compare their effects on CoNV to current therapies such as dexamethasone and bevacizumab. For example, the multitarget TKI, sunitinib, was shown to be 3-fold more effective than bevacizumab [24]. More of this work would help to identify the small molecules that are comparable to or better than current CoNV therapies, and narrow the long list of potential CoNV treatments.

Overall, small molecules have some significant advantages over large molecules in the treatment of CoNV. Biologic pharmaceuticals are drastically more costly than small molecules, therefore resulting in high consumer cost [183]. One reason for this is that analytical characterization of biologics is extremely challenging, requiring combination of numerous methods to ensure their stability and purity [184] whereas small molecules can be structurally verified through high resolution analytical techniques such as NMR spectroscopy [185]. In terms of drug delivery, small molecules can offer greater partitioning across ocular barriers. Oral administration is one of the most common methods of delivery that is noninvasive and patient compliance friendly [186]. However, biologics have poor uptake from the gut and poor absorption across ocular barriers, and therefore oral and other systemic delivery of such large size molecules are typically avoided when oral administration of small molecules can offer more favorable partitioning across ocular barriers [187]. Small molecules can also be formulated as topical eye drops allowing for targeted drug administration to the clinically accessible cornea. Topical administration has the advantage of being noninvasive, with low systemic absorption. Eye drops are easy to administer, and have fairly high compliance rates [188]. Additionally, many of the side effects associated with systemic administration of medications such as cyclosporine A can be avoided with topical eye drop administration [189]. The disadvantage to topical application is that there is decreased bioavailability of the drug by this route. In an attempt to improve local ocular drug delivery, nanoparticle formulations have been tested for EGCG, kaempferol, curcumin, and cyclosporine A with promising results [91,94,112,156]. Therefore, nanoparticles or other advanced formulation of other small molecules may allow for better inhibition of CoNV.

Many of the natural products such as the polyphenols and phytochemicals showed suppression of CoNV; however, their exact antiangiogenic mechanisms and target proteins in CoNV inhibition are not yet known. This is largely due to the non-selectivity of natural products that allows for multitarget disease treatment. For example, the green tea polyphenols, such as EGCG, can inhibit carcinogenesis through inhibition of MAPK, NF-κB, EGFR, insulin-like growth factor, proteasomes, MMPs, urokinase-plasminogen activators, as well as through induction of apoptosis and cell cycle arrest [190]. While further studies are needed to better understand the mechanism of natural products in CoNV, their non-selectivity provides a potential advantage of regulating multiple pathways that may be involved in the complex pathogenesis of CoNV.

Natural products have been studied for their antiangiogenic effects in multiple other disease states. For instance, many antiangiogenic natural products have been tested for their effect on cancer and posterior segment ocular angiogenesis [191,192]. More of those molecules that show potential in treating other angiogenic diseases could be tested in CoNV as well. For example, cremastranone and derivatives are homoisoflavanones that reduce choroidal and retinal neovascularization [193,194,195,196,197] and therefore, may also have activity against CoNV. Moreover, many CoNV studies focus on inhibiting the common pathways associated with vessel proliferation such as VEGF. However, other pathways have been discovered to be involved in angiogenesis, such as heme synthesis via ferrochelatase [198,199,200] and epoxylipid metabolism via soluble epoxide hydrolase [201,202]. Therefore, these could be potential unexplored targets for blocking CoNV.

The small molecules discussed above were mainly studied as monotherapies; however, polypharmacology can have therapeutic benefits. The partial efficacy of bevacizumab is believed to be related to its selectivity for the VEGF pathway. Endothelial cells become less dependent on VEGF as new blood vessels mature due to the recruitment of pericytes [203]. Therefore, this may explain why the multitarget TKI, sunitinib and the combination treatment, sunitinib-hesperetin, have a greater inhibitory effect than bevacizumab alone [24,46]. Multitarget and combination therapies have the advantage of impacting different pathways involved in the pathogenesis of CoNV. Future development of effective CoNV therapies could benefit from combining existing methods with drug discovery of small molecules, repurposed drugs and natural products. Our current knowledge of the application of natural products and numerous small-molecule inhibitors in treating CoNV is limited to in vitro and preclinical in vivo studies but offers exciting potential in future translational studies to investigate their clinical efficacy and safety.

## Figures and Tables

**Figure 1 molecules-25-03468-f001:**
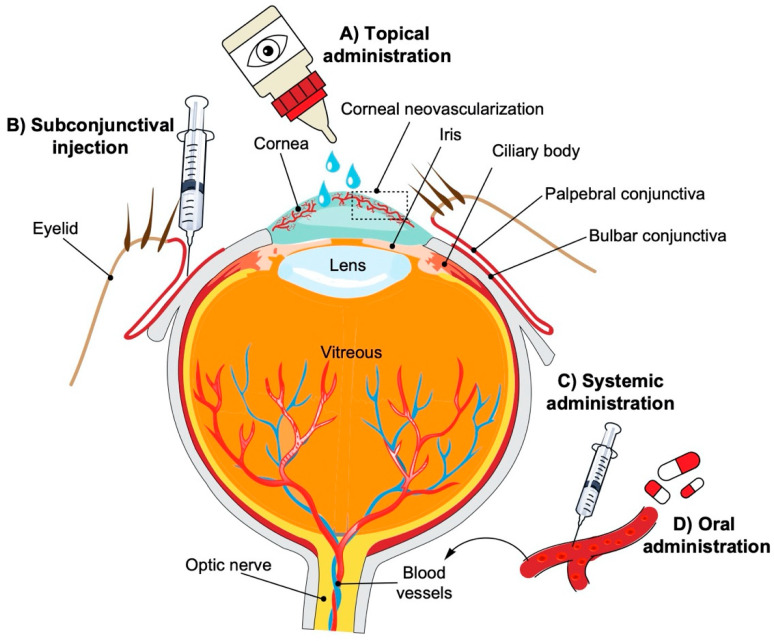
Schematic representation of the human eye and routes of administration for preclinical corneal drug delivery. (**A**) Noninvasive topical administration (eye drops), (**B**) subconjunctival injection given underneath the conjunctiva lining the eyelid, (**C**) systemic administration as intravenous injection, intraperitoneal injection or implanted osmotic pump, and (**D**) oral administration (gavage).

**Table 1 molecules-25-03468-t001:** Tyrosine kinase inhibitors tested in corneal neovascularization (CoNV) models.

Tyrosine Kinase Inhibitor	Source	Mechanism	Routes	Dose	Model	Ref
**Sunitinib** 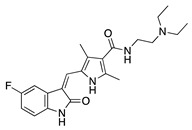	Synthetic	Inhibition of the VEGFR and PDGFR pathways	Oral	40 mg/kg	Murine thermal cauterization	[23]
			Topical	0.5 mg/mL	Rabbit suture model	[24]
			Subconjunctival Topical	0.25 mg/0.1 mL 0.5 mg/mL	Rabbit suture model	[25]
**AG 1296** 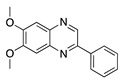	Synthetic	PI3K-RTK inhibition	Systemic via osmotic pump implantation	10 ng/mL	Murine alkali burn model	[27]
**Vatalanib succinate** 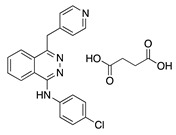	Synthetic	VEGFR inhibition	Oral	75 mg/kg; 2x/day	Murine suture model	[28]
**ZK261991** 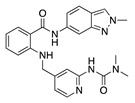	Synthetic	VEGFR inhibition	Oral	50 mg/kg; 2x/day	Murine suture model	[28]
**Sorafenib** 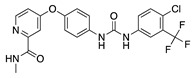	Synthetic	Inhibition of ERK and VEGFR2 phosphorylation	Oral	30 mg/kg; 60 mg/kg	Rat silver-nitrate burn model	[31]
**Semaxanib** 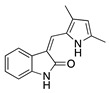	Synthetic	Selective VEGFR2 inhibition	Intraperitoneal	25 mg/kg	Rat silver-nitrate burn model	[33]
**Rivoceranib** 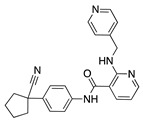	Synthetic	Selective VEGFR2 inhibition	Topical	0.1%; 0.5%	Murine alkali burn model	[36]
**Regorafenib** 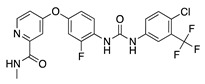	Synthetic	Decreases epithelial and endothelial VEGF levels	Topical	1 mg/mL	Rat alkali burn model	[38]
**Lapatinib** 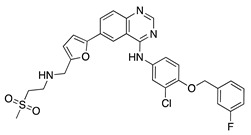	Synthetic	Decreases corneal epithelial and stromal VEGF expression	Oral	50 mg/kg	Rat silver-nitrate burn model	[41]
**Axitinib** 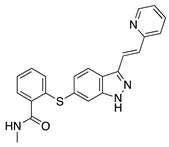	Synthetic	Inhibition of VEGFR2 and PDGFR	Topical	0.02, 0.35, 0.5 mg/mL	Rabbit suture model	[43]
**Dovitinib** 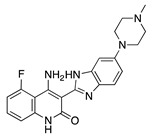	Synthetic	Inhibition of VEGFRs, PDGFR, FGFR-1 and -3,	Topical	5 mg/mL; 2x/day	Rat silver-nitrate burn model	[45]

**Table 2 molecules-25-03468-t002:** Repurposed antimicrobials tested in CoNV models.

Antimicrobial	Source	Mechanism	Routes	Dose	Model	Ref
**Doxycycline** 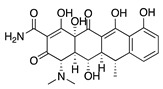	Semisynthetic	MMP inhibition, and modulation of the MMP-independent PI3K/Akt-eNOS pathway	Oral	40 mg/kg	Murine alkali burn model	[49]
Topical	0.5 mg/mL	Murine silver-nitrate model	[50]
**Minocycline** 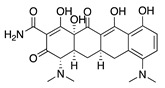	Semisynthetic	Inhibition of MMP and downregulation of the ERK1/2 and Akt pathways	Intraperitoneal	30 mg/kg; 60 mg/kg; 2x/day	Murine alkali burn model	[53]
**Tigecycline** 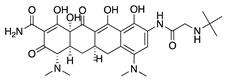	Synthetic derived from minocycline	Unknown	Topical Subconjunctival	1 mg/mL 1 mg/mL	Rat silver-nitrate model	[55]
**Itraconazole** 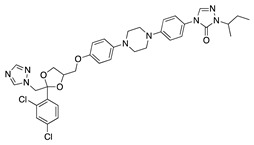	Synthetic	Inhibition of cholesterol biosynthesis	Topical Subconjunctival Intraperitoneal	10 mg/mL 10 mg/mL 19 mg/mL	Rat silver-nitrate model	[57]
**Dihydroartemisinin** 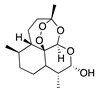	Semisynthetic derivative of artemisinin	Modulation of the ERK1/2 and p38 pathways	Topical	5 mg/L, 10 mg/L, 20 mg/L	Rat suture model	[60]

**Table 3 molecules-25-03468-t003:** Other synthetic small molecules tested in CoNV models.

Molecule	Source	Mechanism	Routes	Dose	Model	Ref
**JSM5562**(Exact structure not reported)	Synthetic	Impairing EC migration, adhesion, and tube formation. Exact mechanism unknown	Systemic via osmotic pump implantation	0.1 mg/mL, 0.5 mg/mL, 2.5 mg/mL	Murine alkali burn model	[62]
**LCB54–0009** 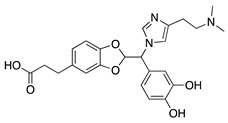	Imidazole-based alkaloid derivative	Regulation of HIF-1α protein stability and HIF-1α/NF-κB redox sensitivity. Inhibits Ang expression and VEGF signaling cascade	Subconjunctival	50 µg/ 20 µL	Rat silver-nitrate burn model	[64]
***N*-acetyl-l-cysteine** 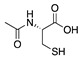	Synthetic	Antioxidant; downregulates VEGF	Intraperitoneal	200 mg/kg	Murine alkali burn model	[66]
**IMD0354** 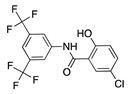	Synthetic	Inhibition of NF-κB through selective blockage of IKK complex, IKK2	Systemic	30 mg/kg	Rat suture model	[68]
**Lanepitant** 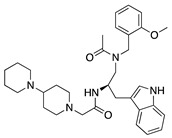	Synthetic	NK1R antagonist	Topical Subconjunctival	0.4, 1.6, 6.4 mg/mL 12.8 mg/mL	Murine alkali burn and suture models	[71]
**SB-328437** 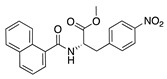	Synthetic	CCR3 antagonist; reduces MCP-1 and -3. Exact mechanism unknown	Topical	125 µg/mL, 250 µg/mL, 500 µg/mL	Murine alkali burn model	[74]
**AMD3100** 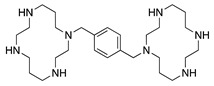	Synthetic	CXCR4 antagonist; Downregulates VEGF expression and inflammation	Subconjunctival Intraperitoneal	5 µL 2.5 mg/kg	Murine alkali burn model	[77]
**33-DFTG** 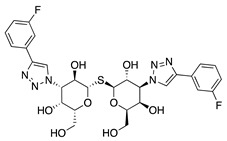	Synthetic	Downregulates VEGF through unknown mechanism	Subconjunctival	50 mM	Murine alkali burn and murine silver-nitrate models	[79]
**TNP-470** 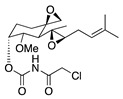	Synthetic analogue of fumagillin	Targets MetAP2	Topical Subconjunctival injection	5 ng/nL; 3x/day 30 mg/kg	Murine alkali burn model	[17]

**Table 4 molecules-25-03468-t004:** Flavonoid polyphenols tested in CoNV models.

Flavonoid	Source	Mechanism	Routes	Dose	Model	Ref
**Epigallocatechin gallate (EGCG)** 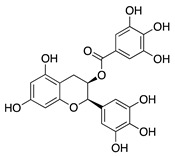	Green tea (*Camellia sinensis*)	Unknown; downregulation of VEGF and COX-2	Topical	0.01 µg/mL 0.1 µg/mL	Rabbit suture model	[89]
			Nanoparticle-mediated delivery via eye drops	30 mg/mL	Murine alkali burn model	[91]
**Kaempferol** 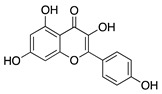	Fruits and vegetables	Unknown; downregulation of MMP and VEGF	Nanoparticle- mediated delivery via eye drops	7.5 µg/mL	Murine silver-nitrate/ potassium model	[94]
**Isoliquiritigenin** 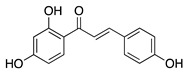	Licorice root (*Glycyrrhiza uralensis*)	Unknown; downregulates VEGF and upregulates PEDF	Topical	0.5, 1, 5, 10, 50 µM	Murine silver-nitrate model	[96]
**Fisetin** 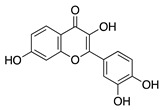	Fruits and vegetables	Unknown	Topical	1.0 mg/mL; 4x/day	Rabbit corneal micropocket b-FGF model	[97]
**Luteolin** 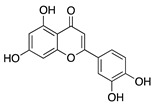	Fruits and vegetables	Unknown	Topical	0.5 mg/mL; 4x/day	Rabbit corneal micropocket b-FGF model	[97]
**Genistein** 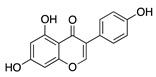	Soybeans	Unknown; downregulates VEGF and TGF-β	Topical	0.5 mg/mL; 4x/day	Rabbit corneal micropocket b-FGF model	[97]
**Naringenin** 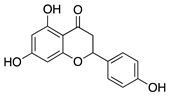	Citrus fruits and vegetables	Unknown; Downregulates NF-κB activity, proangiogenic factors, and reduces production of cytokines IL-1β and IL-6	Topical	0.08, 0.8, 8, 80 µg	Rat alkali burn model	[102]

**Table 5 molecules-25-03468-t005:** Non-flavonoid phytochemicals tested in CoNV models.

Non-Flavonoid Phytochemical	Source	Mechanism	Routes	Dose	Model	Ref
**Curcumin** 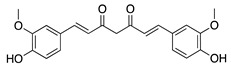	Turmeric (*Curcuma longa*)	Unknown; inhibition of several signal transduction pathways, including NF-κB activation	Topical	40 µM; 2x/day	Rat alkali burn model	[111]
Topical	40, 80, and 160 µM	Rabbit suture model	[110]
Nanoparticle-mediated delivery via eye drops	80 mg	Rat silver-nitrate model	[112]
**Resveratrol** 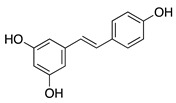	Grapes and other fruits	Unknown; Downregulates FGF-2 and VEGF	Oral	48 mg/kg	Murine FGF-2 and VEGF-micropocket model	[115]
Subconjunctival	10 mg/mL	Rabbit alkali burn model	[116]
**Withaferin A** 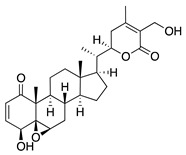	Steroidal lactone (*Withania somnifera*)	Targets and downregulates vimentin	Intraperitoneal	2 mg/kg	Murine de-epithelializ-ation model using wild type and vimentin-null mice	[118]
**Xanthatin** 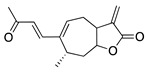	Sesquiterpene lactone (*Xanthium sibiricum*)	Inhibition of the VEGFR2-mediated STAT3/PI3K/Akt signaling pathways	Topical	10 µM; 4x/day	Rat alkali burn model	[123]
**Triptolide** 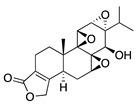	*Tripterygium wilfordii* Hook F	Unknown; downregulates VEGF	Topical	100 nM; 3x/day	Murine alkali burn model	[126]
**Thymoquinone** 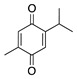	Volatile oil of black seed (*Nigella sativa*)	Unknown; Likely related to antioxidant and anti-inflammatory properties	Topical	0.1%, 0.4%	Rat silver-nitrate model	[131]
**Glycyrrhizin** 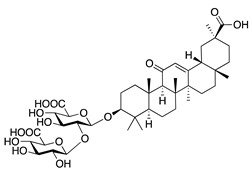	Saponin from licorice root (*Glycyrrhiza glabra*)	Unknown	Topical	1%	Rabbit alkali burn model	[133]

**Table 6 molecules-25-03468-t006:** Immunosuppressants, including macrolides, tested in CoNV models.

Immunosuppressant	Source	Mechanism	Routes	Dose	Model	Ref
**Tetramethylpyrazine** 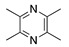	Bioactive component of chuanxiong (*Ligusticum striatum*)	Unknown; Downregulates CXCR-4	Topical	1.5 mg/mL 4x/day	Murine alkali burn model	[137]
**Methotrexate** 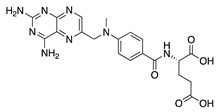	Synthetic	Unknown; Downregulates VEGF and IL-6	Topical Subconjunctival	2 mg/mL, 4 mg/mL 2 mg/mL	Rabbit suture model	[140]
**CC-3052** 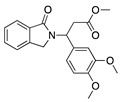	Thalidomide analogue	Unknown; Downregulates VEGF and TNF-α	Topical Subconjunctival	0.25%, 0.5%, and 1% 0.5%	Rabbit suture model	[147]
**DAID** 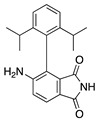	Thalidomide analogue	Unknown; Downregulates VEGF	Topical	0.25%	Murine alkali burn model	[149]
**LASSBio-596** 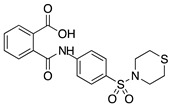	Thalidomide and arylsulfonamide derivative	Unknown	Topical	1%; 3x/day	Rabbit alkali burn model	[152]
**Cyclosporine A** 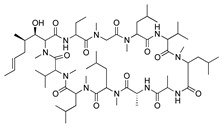	Secondary metabolite of fungal genus *Tolypocladium*	Calcineurin inhibition; downregulates MMP-9, VEGF, and iNOS	Topical Subconjunctival	4% 5 mg/kg	Rat silver-nitrate model	[153]
Topical	0.05%	Rabbit immune-mediated CoNV model	[154]
Nanofibers	0.25 mg/mm^2^	Rabbit alkali burn model	[156]
**Rapamycin** 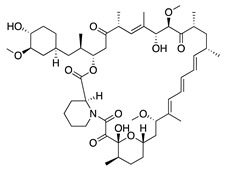	Product of *Streptomyces hygroscopicus*	mTOR inhibition; downregulates VEGF, TNF-α, TGF-β, IL-6, and Substance P	Topical Intraperitoneal	1 mg/mL 0.2 mg/kg	Murine alkali burn model	[160]
Intraperitoneal	2 mg/kg; 1x/day	Murine alkali burn model	[159]
**Tacrolimus** 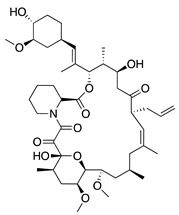	Product of *Streptomyces tsukubaensis*	Calcineurin inhibition; downregulates VEGF, TNF-α, IL-1β, and MCP-1	Topical Subconjunctival	5 mg/5 mL 0.25 mg/ 0.05 mL	Rabbit suture model	[164]

**Table 7 molecules-25-03468-t007:** Vitamins and photoactivatable small molecules tested in CoNV models.

Vitamin/Photoactivatable Compound	Source	Mechanism	Routes	Dose	Model	Ref
**Ascorbic acid** 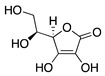	Diet	Unknown; Downregulation of VEGF and MMP-9	Topical	0.5, 1, 10 mg/mL	Rabbit suture model	[167]
**Riboflavin** 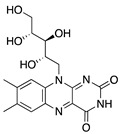	Diet	Induction of apoptosis in vascular ECs; downregulation of macrophages and CD45+ cells	Topical riboflavin followed by UVA exposure	0.1%	Murine suture model	[171]
**Verteporfin** 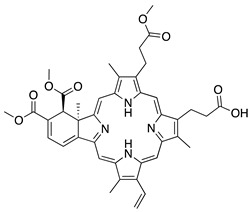	Synthetic	Suppressed blood vessels and lymphatic vessels	Intravenous followed by light exposure	6 mg/m^2^	Murine suture model	[174]
**1α,25-dihydroxyvitamin D_3_** 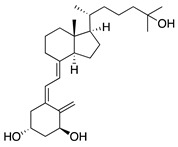	Diet	Inhibited migration of Langerhans cells into cornea	Topical	10^−7^ M, 10^−8^ M, and 10^−9^ M	Murine suture model	[177]

**Table 8 molecules-25-03468-t008:** HDAC inhibitors tested in CoNV models.

HDAC Inhibitor	Source	Mechanism	Routes	Dose	Model	Ref
**Largazole** 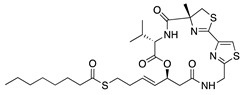	Macrocyclic depsipeptide from marine cyanobacterium *Symploca* species	Class I HDAC inhibition; downregulates VEGF, b-FGF, TGF-β1, and EGF; Upregulates Tsp-1, Tsp-2, and ADAMTS-1	Topical	5 µL; 2x/day	Murine alkali burn model	[180]
**Vorinostat** 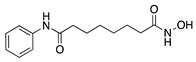	Synthetic	HDAC inhibition; targets unknown	Topical	10 µM; 3x/day	Murine alkali burn model	[182]

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
