# Peer review of "Pharmacological Potential of Small Molecules for Treating Corneal Neovascularization"

_molecules, 2020, doi:10.3390/molecules25153468_

Round 1
Reviewer 1 Report
The manuscript entitled “Pharmacological Potential of Small Molecules for Treating Corneal Neovascularization” discusses the effects of the repurposed antimicrobials, natural source-derived flavonoid, non-flavonoid phytochemicals, immunosuppressants, vitamins as the inhibitors of the vascular endothelial growth factor (VEGF) receptor and other tyrosine kinases and histone deacetylase. These small molecules induce anti-angiogenic and anti-inflammatory effects through inhibition of VEGF, NF-kappa-beta, and other growth factor receptor pathways. Taken together, the manuscript summarizes the anti-angiogenic effect of the small molecules in the treatment of corneal neovascularization. The manuscript requires minor revision.
The below revisions are recommended:
It is better to categorize Tacrolimus, Rapamycin, Cyclosporine A under a separate sub-heading “Macrolides” rather than “Immunosuppressants.”
Lines# 551-553: Thalidomide has two enantiomers. Both are NOT teratogen. The ®-Thalidomide is a teratogen. It should be mentioned in the text.
Line#420: “Curcumin is a non-flavonoid polyphenol…” It might be better to say, “Curcumin, the major curcuminoid polyphenol…..”
Uniformity (font and size) should be mentioned throughout the manuscript including the molecular structures.
The manuscript requires a thorough editing of punctuation. For example, the in vitro and in vivo should be italics. The authors are encouraged to check the journal IFA.
Author Response
Thank you for reviewing the manuscript and offering recommended revisions. Please note the changes made to the manuscript below.
It is better to categorize Tacrolimus, Rapamycin, Cyclosporine A under a separate sub-heading “Macrolides” rather than “Immunosuppressants.”
In response to this suggestion, we have created a subsection for Macrolides (Cyclosporin A, Rapamycin, and Tacrolimus) (Line 563).
Lines# 551-553: Thalidomide has two enantiomers. Both are NOT teratogen. The ®-Thalidomide is a teratogen. It should be mentioned in the text.
We have now added more information added about the (S) & (R) thalidomide enantiomers and the teratogenic effect of the (S) enantiomer (Lines 539-546).
Line#420: “Curcumin is a non-flavonoid polyphenol…” It might be better to say, “Curcumin, the major curcuminoid polyphenol…..”
We have made this change as suggested (Line 444).
Uniformity (font and size) should be mentioned throughout the manuscript including the molecular structures.
We have made the font (Palatino Linotype) and font size (10 pt) uniform throughout the manuscript (except references and back matter, consistent with the journal template), including converting all Greek characters to Palatino Linotype. In addition, we have adjusted the structures in the tables to be more uniform as well.
The manuscript requires a thorough editing of punctuation. For example, the in vitro and in vivoshould be italics. The authors are encouraged to check the journal IFA.
We have checked the entire manuscript against the journal guidelines. In particular, we have now ensured that "in vitro" and "in vivo" are italicized throughout.
Reviewer 2 Report
Dear Dr. Corson
The manuscript “Pharmacological Potential of Small Molecules for 3 Treating Corneal Neovascularization” is a comprehensive review on synthetic and natural molecules with potential in corneal neovascularisation. Similar reviews on that topic are available, but in my opinion, this manuscript stands out through extend and deepness of analysis. The review is mainly research based oriented, targeting the molecular mechanism of synthetic and natural molecules. The review include a significant and relevant literature, many of them recently published. The style is elegant and easily comprehensible. Therefore, I appreciate the manuscript may be published in the actual form.
Author Response
We thank the reviewer for their favorable comments.